# Possible Applications of Fecal Microbiota Transplantation in the Pediatric Population: A Systematic Review

**DOI:** 10.3390/biomedicines13061393

**Published:** 2025-06-06

**Authors:** Ewa A. Bieganska, Przemyslaw Kosinski, Marek Wolski

**Affiliations:** 1Department of Pediatric Surgery, Medical University of Warsaw, 02-091 Warsaw, Poland; marek.wolski@wum.edu.pl; 2Department of Obstetrics, Perinatology and Gynecology, Medical University of Warsaw, 02-091 Warsaw, Poland; przemyslaw.kosinski@wum.edu.pl

**Keywords:** fecal microbiota transplantation, gut microbiome, gut dysbiosis, systematic review, pediatrics, *Clostridioides difficile* infection, inflammatory bowel disease, autism spectrum disorders

## Abstract

**Background**: The potential therapeutic role of fecal microbiota transplantation (FMT) in various diseases has been thoroughly studied over the last few decades. However, the majority of studies focus on the adult population, therefore, conclusions regarding the application of FMT in the pediatric population are much less clear. This systematic review aims to summarize the research conducted so far on the efficacy and safety of FMT in the pediatric population, assess the quality of the evidence of its effectiveness, and outline the most promising areas for future research. **Methods**: We performed a systematic literature search from the index date to 8 June 2024 on the Embase, PubMed, and Web of Science databases. One author screened the resulting 121 articles. Eventually, 35 eligible studies that reported FMT use in seven different diseases were identified. **Results**: All of the studies assessed FMT as a safe procedure without many serious adverse effects. The best-documented application, which is the only one recommended in official guidelines, is recurrent *Clostridioides difficile* infection. Other disease entities in which the use of FMT has been studied with good clinical effects are inflammatory bowel disease, allergic colitis, autism, Tourette syndrome, and colonization with multi-drug-resistant organisms. However, it should be noted that the majority of studies are cohort and case-control studies, without randomization, which translates into low evidence quality. In one randomized, controlled trial focusing on the effect of FMT on weight loss in obese individuals, a lack of effect was found. **Conclusions**: While FMT and subsequent iterations of gut microbiota-targeted interventions hold promising therapeutic potential for various disease entities in the pediatric population, the current evidence behind this conclusion is of low quality. Based on current studies, these methods appear to be both effective and safe. However, further randomized clinical trials are necessary, especially within the pediatric population, for which such studies remain scarce.

## 1. Introduction

The gut microbiome and its impact on human health have been the subject of an increasing number of studies over the past two decades. The influence of dysbiosis on diseases, such as inflammatory bowel diseases, irritable bowel syndrome, neurological disorders such as Parkinson’s disease and Alzheimer’s disease, and even on the effectiveness of oncological therapies, appears to be increasingly well-proven [1,2,3,4]. Recently, attention has also been paid to the very significant impact of microbiota on the development of the human nervous system [5,6]. The question arises whether the change in the composition of the microbiota to a more ‘physiological’ one could be an effective therapeutic method. One of the therapies with the potential to alter microbiota composition is fecal microbiota transplantation (FMT). The FMT procedure involves transferring a bacterial suspension prepared from the feces of a healthy donor into the recipient’s digestive tract. This preparation contains all the microorganisms, metabolites, and proteins that occur naturally in a healthy donor’s gut. Unlike probiotics, which are selective and strain-dependent, the microbiota contains the full antigenic and metabolic repertoire. The possible routes of administration are via the upper gastrointestinal tract (e.g., oral capsules or the administration of a suspension into the duodenum during gastroscopy) or via the lower gastrointestinal tract (e.g., the administration of a suspension during colonoscopy or deep rectal infusion). Colonoscopic administration is the most well-documented method in the literature, whereas upper gastrointestinal administration is associated with a higher incidence of reported side effects [7,8,9].

FMT treatment has been proven to be highly effective in restoring intestinal eubiosis. To date, the only indication for FMT that is already considered in official standard guidelines is recurrent *Clostridioides difficile* infection in both adults and children. The effectiveness of this method has been demonstrated in numerous studies [10,11,12]. However, substantial evidence from various randomized clinical trials supports the potential effectiveness of FMT in other conditions, though this has not yet been reflected in official guidelines. Studies on the efficacy of achieving remission in inflammatory bowel disease and eradicating antibiotic-resistant pathogens are at an advanced stage. Additionally, many clinical trials are underway to investigate newer applications of FMT. Some of the more promising areas of research include hepatic encephalopathy, irritable bowel syndrome, metabolic syndrome, and a wide range of neurological and psychiatric disorders [9,13]. Most studies, however, focus on adult patient populations, and therefore, conclusions regarding the efficacy and safety of FMT in the pediatric population are currently much less clear. The therapeutic successes of FMT in the adult population and increasing research pointing to the potential effectiveness of microbiota-targeted therapies in diseases predominantly affecting children create a growing need for high-quality clinical trials specifically for this population. Diseases that have recently been the focus of increasing research include inflammatory bowel diseases (IBD), other autoimmune diseases, obesity, and autism spectrum disorders.

This systematic review aims to summarize the research conducted so far on the efficacy of FMT in the pediatric population, assess the quality of the evidence on its effectiveness, summarize the considerations regarding the safety of this method in children, and outline the most promising areas for future research.

## 2. Methods

Literature review

This study is a systematic review of experimental and observational studies, conducted according to the Preferred Reporting Items for Systematic Reviews and Meta-Analyses (PRISMA) guidelines, and includes all elements of the PRISMA checklist that could be completed.

We performed a systematic literature search from the index date to 8 June 2024 on Embase, PubMed, and Web of Science databases, using the search algorithms included in Appendix A.

EndnoteX9 was used to retrieve titles and abstracts for initial investigations and to remove duplicates. The articles were then reviewed initially by title and abstract, then by full text. The screening was performed by one author. Additional studies were found by cross-referencing. The study selection process according to the Preferred Reporting Items for Systematic Reviews and Meta-Analyses (PRISMA) statement is detailed in Figure 1 [14].

For the ongoing clinical trials review, the ClinicalTrials.gov website was consulted [15].

Eligible studies were selected and assigned to different groups according to an indication of the FMT, and then they were analyzed for the type of study, reported efficacy and safety, and quality of evidence (according to the National Health and Medical Research Council (NHMRC) evidence hierarchy).

Exclusion and inclusion criteria

Studies that reported on the clinical efficacy and/or safety of fecal microbiota transplantation in pediatric (<18 years old) or adolescent (<21 years old) patients and were written in English were included. Clinical trials, case series, and cohort studies were included. In the case of diseases for which it had a well-proven efficacy, official guidelines, systematic reviews and meta-analyses were also consulted and included. We excluded experimental pre-clinical studies, narrative reviews, and opinions.

Language support and text translation

To ensure the high quality and accuracy of text translation, the AI-based tool ChatGPT–3.5, developed by OpenAI, was used. It was used as a language support tool to translate text fragments from the native language of the authors to English. All final versions of the text were verified for content and language correctness by the author.

## 3. Results

We identified 35 studies eligible for the systematic review, regarding the use of FMT in the following diseases in the pediatric patient population (an overall summary of studies’ results is in Table 1):Recurrent *Clostridioides difficile* infection (rCDI), distinguishing between the following populations: CDI in patients with cancer and CDI in patients with IBDIBD, distinguishing between the following populations: those with Ulcerative colitis (UC) and those with Crohn’s disease (CD)Autism spectrum disorders (ASD)Allergic colitisObesityTourette syndromeMulti-drug-resistant organism (MDRO) decolonization

### 3.1. Clostridioides Difficile Infection

*Clostridioides* (previously *Clostridium*) *difficile* infection is one of the common causes of antibiotic-related diarrhea and, at the same time, one of the most common healthcare-related infections. In children, unlike in adults, this infection is much more often community-associated, and a fulminant course is less frequently observed. However, similarly to adults, this infection is characterized by an increasing incidence and a high risk of recurrence, reaching up to 20–30% [11,16]. Recurrence is defined as a new onset of CDI symptoms with a positive *C. difficile*-specific laboratory test following an incident episode in the previous 2 to 8 weeks. FMT as a method of treating recurrent infection has been recognized in official guidelines for several years now. In children, however, the evidence for the effectiveness of this method is of lower quality, and concerns about administering microbiota at an age when the intestinal microbiome is still naturally forming are greater.

For the purposes of this review, due to the substantial amount of evidence, the presence of the indication in official guidelines, and the availability of high-quality systematic reviews, we decided to summarize the available meta-analyses. There were two identified publications which included the majority of available studies regarding the effectiveness and safety of FMT in CDI or rCDI in pediatric populations.

In a systematic review and meta-analysis from 2022 performed by Tun KM et al., 904 pediatric patients, who received FMT for CDI or/and rCDI, were identified. The data were collected from 14 studies, none of which were randomized, controlled trials. The gross success rate was 81.86% (740/904), while the overall failure rate was 18.14% (164/904). The calculated pooled rate of clinical success of FMT in the overall cohort was 86%.

The most common concomitant disease in this study was IBD, which was reported in 337 patients. The overall success rate of FMT for CDI in IBD patients was 75.33% (223/296), while the failure rate was 24.66% (73/296).

Regarding the safety of the procedure, 38 serious adverse events (SAEs) in 36 patients and 47 adverse events (AEs) in 45 patients were reported. Among SAEs, IBD flares and IBD-related surgery were listed as the most common. Among AEs, diarrhea and abdominal pain were the most common. There were no deaths attributable to FMT [17].

In a systematic review and meta-analysis performed by Tariq R. et al. in 2023, in which the authors focused on CDI in patients with IBD, a total of six studies including 141 patients were identified [18]. All of the studies were observational. Of the identified patients, 106 had CDI resolution after the first FMT with a pooled cure rate of 78%. The overall pooled cure rate with single and multiple FMTs was 77%. The overall cure rate for CDI with multiple FMTs was not higher in the pediatric population. It is also worth noting that 10 of 31 patients experienced an improvement in IBD symptoms after FMT, with a pooled rate of 32.2%. Nevertheless, the risk of IBD flares after FMT was also emphasized, as they occurred in a total of 13 of 120 pediatric patients, with a pooled rate of 10.8%.

Another group that is worth considering separately from the CDI group is oncology patients. In the official guidelines for the Management of *Clostridium difficile* Infection in Children and Adolescents with Cancer and Pediatric Hematopoietic Stem-Cell Transplantation Recipients developed by experts from the American Society of Clinical Oncology in 2018, a strong recommendation against the use of FMT was made. The rationale behind this decision was the lack of randomized data, especially in immunocompromised patients, and challenges related to the mode of administration. However, the need to include the pediatric population in further clinical trials to assess FMT safety and clinical effectiveness was emphasized. In the 2024 update of the guidelines, despite the inclusion of seven new clinical trials in the review, the recommendation against the use of FMT in this patient population was maintained. The decision was again justified by the small amount and low quality of evidence in this population and the lack of data specifically in neutropenic patients [19,20]. It is worth mentioning, however, that there are cases in the literature in which FMT was performed in oncological, immunosuppressed patients, and in most cases, no serious side effects were reported, and FMT itself had a good therapeutic effect [21,22]. However, this is very low-quality evidence.

In summary, the use of FMT to treat CDI in children demonstrates high efficacy and an acceptable safety profile. However, special attention should be paid to patients with IBD, as there is a significant risk of disease exacerbation. FMT is still not recommended for oncology patients due to insufficient scientific evidence, although case reports suggest its potential efficacy in certain situations.

### 3.2. Inflammatory Bowel Disease

Inflammatory bowel diseases, including ulcerative colitis and Crohn’s disease, are chronic inflammatory disorders of the gastrointestinal tract caused by a dysregulated mucosal immune response to the intestinal microflora. IBDs begin most commonly during adolescence and young adulthood, with approximately 25% of patients presenting symptoms before the age of 20 years. In children, 4% of IBDs are present before the age of 5 and 18% before the age of 10 [23]. Although genetic predispositions play an important role in its pathogenesis, an increasing number of studies point to dysbiosis as a strong contributing factor [24,25]. For this reason, the effectiveness of FMT in the treatment of these diseases has begun to be investigated, especially since it could be an effective alternative to immunosuppressive therapy, which is burdened with many side effects, particularly in patients of a young age.

We identified seven clinical trials focused on the safety and efficacy of fecal microbiota transplantation in the treatment of inflammatory bowel diseases in children. While the majority of clinical reports are of low quality (Level III-3 to Level IV), there have been a few randomized clinical trials conducted on the effectiveness of FMT in both Crohn’s disease and ulcerative colitis. Two of them are PediCRaFT and PediFETCH (their IDs from the ClinicalTrials website are, respectively, NCT03378167 and NCT02487238). While the PediCRaFT trial has been completed quite recently and the results are still not published, apart from initial reports regarding problems in recruiting an appropriate number of patients, the results from the PediFETCH trial are available [26,27]. Although the researchers did not reach their primary feasibility outcome of achieving recruitment targets, they did reach the composite clinical endpoint (an improvement in pediatric UC activity index, C-reactive protein, or fecal calprotectin) in 92% of patients assigned to FMT vs. 50% assigned to a placebo at week 6. Furthermore, at 12 months, 75% had maintained a clinical response. In the conclusions of their work, the authors drew attention to the difficulties associated with recruiting patients and compliance [28,29].

In another study regarding ulcerative colitis by Le J. et al., significant correlations between improved clinical outcomes and marked shifts in the gut microbiome within both the treatment and placebo groups were reported. At the 4-week mark, clinical improvement or remission was observed in 46% of patients in the treatment group and in 82% of patients in the placebo group [30].

In other studies, regarding mostly patients with UC, characterized by having lower-quality evidence and smaller sample sizes, as well as usually focusing on patients who do not respond to recognized lines of treatment, good treatment effects have been reported, with even more than 70% of patients having a clinical response to therapy. Most reports have assessed the procedure as safe, with rare serious complications [31,32,33,34,35].

In the systematic review by Imdad A. et al. from 2023 regarding mostly adult patients with inflammatory bowel disease, it was found that while FMT may increase the proportion of people with active UC who achieve clinical and endoscopic remission, the evidence is very uncertain about the use of FMT for the maintenance of remission in people with UC, as well as the induction and maintenance of remission in people with CD [36]. Similar conclusions for pediatric patients are currently much more difficult to draw, due to the limited number of high-quality clinical studies, so the current evidence is based mostly on case series. However, the systematic review by Hsu M. et al., 2023, in which the authors focused on IBD in the pediatric population, reported a clinical response rate one month after FMT that reached almost 60%. The calculated pooled rate of adverse events was 29%, and the calculated pooled rate of serious adverse events was 10% [37].

In conclusion, the use of FMT in IBD in children offers hope as a potentially effective alternative to immunosuppression with relatively low rates of side effects. However, clinical data are still limited. The results of clinical trials suggest the possibility of clinical improvement, even in patients who do not respond to standard treatment, particularly in ulcerative colitis. However, significant difficulties in patient recruitment and compliance prevent us from drawing clear conclusions.

### 3.3. Autism

Autism spectrum disorder (ASD) is used to describe a heterogeneous set of neurodevelopmental conditions, characterized by a range of impairments in social communication, social interaction, and repetitive behaviors. The symptoms are observed as early as in 1 year of age and are usually associated with lifelong challenges for both the affected individuals and their families. Apart from psychological and behavioral symptoms, a higher prevalence of gastrointestinal symptoms, sleep disturbances, and immune dysfunction is reported [38]. Although it is believed that autism has a particularly large genetic contribution, its pathophysiology is still not fully understood, and one of the proposed theories highlights the possible influence of gut microbiota and the gut–brain axis on ASD [6,39,40].

In our systematic review, we identified six clinical trials examining the impact of FMT on psychological, behavioral, and gastrointestinal symptoms in autism. Four of them were open-label clinical trials without randomization, one was a retrospective study (both Level III-3 according to the NHMRC evidence level) [41,42,43,44], and one was an open-label, randomized waitlist-controlled trial (Level II), although only an abstract form of the last one is currently available [45]. Two of the publications by Kang et al. in 2017 and 2019 concerned the same study group, in which the effectiveness assessment was repeated after 2 years [41,42].

All of the included studies reported a significant improvement in both GI and autism symptoms. The average Gastrointestinal Symptom Rating Scale (GSRS) score, which is typically utilized for the assessment of GI symptoms, dropped between 35 and 77% in all of the studies. What is more, in the group assessed 2 years after the intervention, the improvement was maintained [41].

The scores on CARS, which rates core ASD symptoms, were decreased by 10% to 22% after the treatment. In the long-term observation period, the score continued to drop, reaching a 47% decrease from the baseline. A positive difference was also noted in the ABC score, which evaluates irritability, hyperactivity, lethargy, stereotypy, and aberrant speech (up to a 35% decrease). In one study, in which the VABS-II score was assessed, which evaluates adaptive behaviors such as communication, daily living skills, and socialization, an increase in the average developmental age was reported [41,42].

In the systematic review and meta-analysis performed by Zhang et al., 2023, the authors confirmed the potential of FMT as a therapy for alleviating symptoms of ASD. A reduction in the CARS, ABC, and SRS scores after the FMT was observed. However, the need for rigorously designed randomized double-blind placebo-controlled trials was highlighted to establish the safety and efficacy of FMT as a treatment for ASD [46].

In summary, the available data suggest that FMT can provide significant benefits when treating the gastrointestinal symptoms and core symptoms of ASD, including improvements in adaptive behavior, with no significant side effects reported. However, these results are based primarily on studies of low methodological quality, and the apparent placebo effect in some studies limits the strength of the conclusions that can be drawn.

### 3.4. Other Conditions

For the remainder of the aforementioned diseases, our systematic review summarizes studies examining the effectiveness of FMT. Their results are summarized in Table 2.

### 3.5. Possible Future Indications

Some of the most promising research directions in recent years include its use in treating autoimmune diseases (such as type I diabetes and Hashimoto’s disease), metabolic disorders, and neurological diseases (such as multiple sclerosis and neurodegenerative diseases). Studies have also noted the potential anti-inflammatory and immunomodulatory effects of FMT, which may support therapies for systemic infections or cancers [50,51,52,53,54].

Many of these conditions affect patients from a young age, creating a need for research in the pediatric population as well.

There are currently 77 registered clinical trials on ClinicalTrials.gov [15] regarding the use of FMT in the pediatric population. A summary of them is shown in Table 3.

## 4. Discussion

The therapeutic use of fecal microbiota transplantation (FMT) has been a widely researched topic in recent years, generating significant hope for therapy. With increasing scientific reports uncovering new connections between gut dysbiosis and various disease entities, the volume and diversity of conducted research have grown. Currently, there are over 500 registered clinical trials on ClinicalTrials.gov utilizing fecal microbiota transplantation. It is worth noting that some of these studies focus on diseases that mainly affect children. These diseases are associated with a significant burden for patients, who face a lifetime of illness, as well as for their families, who are involved in the treatment process. This trend also extends to pediatric and adolescent patients. The efficacy of FMT in recurrent *Clostridioides difficile* infection is well documented and reflected in official guidelines for both adults and children. Other conditions for which this intervention is gathering substantial evidence include inflammatory bowel diseases (IBD). Although a greater efficacy is observed in ulcerative colitis, the results for Crohn’s disease remain less clear and involve a smaller patient cohort. Studies concerning autism spectrum disorder (ASD) also report promising results, confirming the therapeutic impact of FMT.

A particularly relevant aspect for the pediatric population is the safety of the method, which is positively assessed across all studies. Although serious complications may occur, particularly in immunocompromised patients and especially with FMT administration via the upper gastrointestinal tract, these do not exceed 1.5% of the studied cases [55,56]. Other safety considerations specific to this age group refer to the uncertain influence of microbiota changes in the early years when the microbiome is still evolving.

Despite initial high hopes for microbiome-targeted interventions, scientific research faces several obstacles. It is important to note that most studies are characterized by low-quality evidence and a very limited number of randomized clinical trials. This can be caused by a range of difficulties encountered by researchers when working with pediatric patients, such as problems recruiting a sufficient number of patients or finding an appropriate study group. Another issue is the application of fecal microbiota transplantation (FMT), which requires proper patient preparation and frequent visits to the facility, leading to a relatively high dropout rate.

Nevertheless, current studies show trends toward facilitating the FMT administration process, such as the use of FMT capsules. Efforts are also being made to optimize other delivery routes and preparatory protocols. Additionally, attempts are being made to match donors to recipients and their specific disease conditions individually, as it appears that materials from different donors may have varying levels of effectiveness [57,58].

The necessity of conducting randomized control trials is consistently emphasized for conditions such as inflammatory bowel diseases, neurological disorders, and autoimmune diseases [59,60,61]. One of the relatively recently discovered directions for FMT is its application in patients with necrotizing enterocolitis (NEC). The experimental studies conducted so far report very promising results both in the prevention of and possible therapy for NEC. The authors of this review paper are currently conducting a clinical trial for this indication.

This review paper may help clinicians involved in pediatric medicine to understand the current state of research into the use of FMT in children. This is particularly important given that this procedure is proven to have positive effects on an increasing number of diseases, some of which affect children in particular. The most interesting current indications, which may soon enter wider clinical use, are autism spectrum disorders, MDRO eradication, and inflammatory diseases of newborns. Given its relatively low cost and few side effects, FMT could become a very significant component of many therapeutic options in the years to come.

Nevertheless, this systematic review has several limitations. Firstly, the aforementioned heterogeneity among the included studies and variations in the outcome measures used may have impacted the consistency of the findings and the comparability of the studies, particularly given that the majority of the studies are characterized by low-quality evidence. Secondly, the small number of randomized controlled trials (RCTs) reduces the robustness of the conclusions. Third, restricting the literature search to a limited number of databases is a limitation of this systematic review. Consequently, relevant studies published in other sources may have been overlooked, which could affect the review’s comprehensiveness.

In summary, FMT and subsequent iterations of gut microbiota-targeted interventions hold promising therapeutic potential for various disease entities. Based on current studies, these methods appear to be both effective and safe. However, further randomized clinical trials are necessary, especially within the pediatric population, for which such studies remain scarce.

## Figures and Tables

**Figure 1 biomedicines-13-01393-f001:**
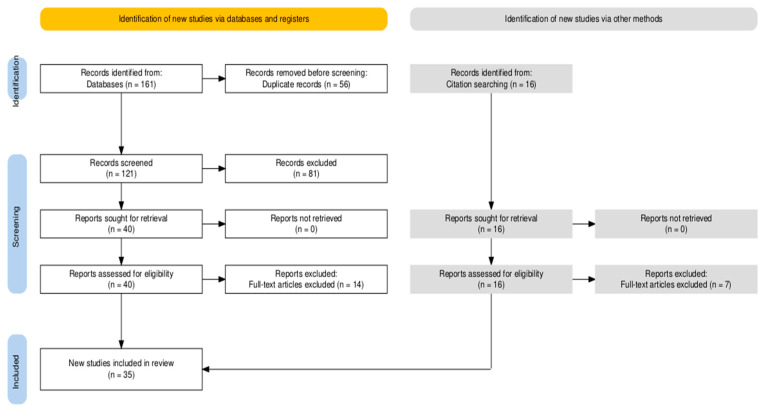
PRISMA flow diagram.

**Table 1 biomedicines-13-01393-t001:** Overall summary of studies.

Disease	Level of Evidence *	Number of Included Studies	Results
Clostridioides difficile infection (CDI)	Systematic reviews of Level III studies (cohort and case-control studies); Level III-2	10	Successful, recommended in official guidelines, although further evidence is needed
Ulcerative colitis (UC) and Crohn’s disease (CD)	Single randomized clinical trials; Level II cohort and case-control studies; Level III-3 case series; Level IV	14	Successful and safe, although evidence is of overall low quality and in some cases conflicting
Autism spectrum disorders (ASD)	Systematic reviews of Level III studies (cohort and case-control studies); Level III-2	7	Successful in alleviating both ASD and GI symptoms; randomized control trials are needed
Allergic colitis	One clinical trial, open-label, no controls; Level III-3	1	Successful and considered as safe
Multi-drug-resistant organism (MDRO) eradication in cancer patients	Case series; Level IV	1	Successful decolonization; Considered as safe
Obesity	One randomized control trial; Level II	1	No effect of FMT on weight loss. Post-hoc analyses indicated a resolution of undiagnosed metabolic syndrome. No serious adverse events.
Tourette syndrome	Preliminary study, case series; Level IV	1	Successful in achieving clinical response. No serious adverse events.

* According to the NHMRC evidence hierarchy.

**Table 2 biomedicines-13-01393-t002:** Detailed summary of studies.

Disease	Level of Study	Study Characteristics and Results
Allergic colitis	Open-label, prospective, single-center trial; Level III-3 [47]	Nineteen infants who did not respond to standard therapy and could not adhere to it. After FMT treatment, allergic colitis symptoms in 17 infants were relieved within 2 days, and no relapse was observed in the next 15 months. Beyond one patient who suffered from eczema, no other adverse event was recorded during FMT or the follow-up.
Tourette syndrome	Open-label, prospective, single-center trial; Level III-3 [48]	Five patients who had been diagnosed for more than one year had a persistently high level of tic severity and had a relapse or were intolerant to regular medications for tic disorders. At week 8 after FMT, 4/5 patients achieved a clinical improvement. The combined motor tic and vocal tic scores of the 4 patients decreased with a range of 7–35. During the FMT process and follow-up period, no patients experienced any obvious adverse events.
Decolonization of multi-drug-resistant bacteria before allogeneic hematopoietic stem cell transplantation	Case series; Level IV [22]	Five patients colonized by MDR bacteria underwent FMT before HSCT. Multi-drug-resistant decolonization was achieved within one week in 4 of 5 patients. At the 1-month follow-up, 4 previously decolonized patients switched to a new colonization status (from the same pathogen identified before FMT), and one patient who was still colonized after FMT achieved decolonization. Four patients did not experience serious AE, while one suffered from an episode of sepsis (from the same pathogen for which he received FMT) 17 days after the procedure. Repeated FMT could increase the chances of durable MDR decolonization.
Obesity	Randomized, double-masked, placebo-controlled trial; Level II [49]	Eighty-seven adolescents aged 14 to 18 years with a body mass index of 30 or more. There was no effect of FMT on BMI standard deviation score at 6 weeks. Reductions in android-to-gynoid-fat ratio in the FMT vs. placebo group were observed. There were no observed effects on insulin sensitivity, liver function, lipid profile, inflammatory markers, blood pressure, total body fat percentage, gut health, or health-related quality of life. In post-hoc exploratory analyses among participants with metabolic syndrome at baseline, FMT led to greater resolution of this condition compared with placebo. There were no serious adverse events recorded throughout the trial.

Abbreviations used in Table 2: FMT—fecal microbiota transplantation; MDR—multi-drug-resistant; HSCT—hematopoietic stem cell transplantation; AE—adverse event; BMI—body-mass index.

**Table 3 biomedicines-13-01393-t003:** Currently registered trials regarding FMT and pediatric patients.

Disease	Number of Trials
Graft-versus-host disease	6
Crohn’s disease	7
Inflammatory bowel disease (in general)	7
Ulcerative colitis	9
*Clostridioides difficile* infection	12
Functional gastrointestinal disorders	3
Autism spectrum disorder	7
Antibiotic-resistant bacteria decolonization	4
Irritable bowel syndrome	1
Epilepsy	1
Metastatic mesothelioma	1
Severe motility disorder	1
Necrotizing enterocolitis	1
Small intestinal bacterial overgrowth	1
Malnutrition	1
Overall safety and efficacy of FMT	2
Attention-deficit hyperactivity disorder	1
Rhinitis	1
Intestinal microbiome of the newborn after Cesarean section	1
Chronic granulomatous disease-associated colitis	1
Chronic kidney disease	1

## Data Availability

The data supporting this study’s findings are available in the following databases: PubMed, Embase, and Web of Science. These data were derived from the aforementioned resources using the algorithm provided in Appendix A.

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
