# Peer review of "Possible Applications of Fecal Microbiota Transplantation in the Pediatric Population: A Systematic Review"

_biomedicines, 2025, doi:10.3390/biomedicines13061393_

Round 1

Reviewer 1 Report

Comments and Suggestions for Authors

The authors conducted a systematic review on the applications of fecal microbiota transplantation in the pediatric population. The study is clear and comprehensive. Minor modifications are suggested for clarity and improved readability. (See the file attached).

- The objective is worded differently in the abstract and the introduction. It is recommended to complete and unify the objective.
- The names of the microorganisms are italicized, and the bacterial genus is always capitalized.
- The methodological design of the study is missing from the Materials and Methods section (quantitative, observational, cross-sectional, etc.).
- All abbreviations must be described in the text the first time they are mentioned.
- The tables are independent; therefore, all study information and abbreviations must be included in each table.
- The authors may include limitations of the study.
- The references are not formatted uniformly and are not consistent with the journal's style. Review and correction are recommended.

Author Response

We would like to express our sincere gratitude to the reviewers for their thoughtful comments, insightful suggestions, and constructive criticism. Their valuable feedback has significantly contributed to improving the quality and clarity of our manuscript. We have carefully addressed each point raised, and we believe that the revisions made have strengthened the work and enhanced its potential contribution to the medical community. We appreciate the opportunity to refine our study and hope that the revised version meets the expectations of the reviewers and editors.

To make the revised manuscript clearer, the passages that have been amended have been highlighted in yellow.

Responses to the comments:

  • The objective is worded differently in the abstract and the introduction. It is recommended to complete and unify the objective – I have revised the manuscript to complete and unify the wording of the objective throughout the abstract and the introduction, ensuring consistency and clarity. Your suggestion is greatly appreciated.
  • The names of the microorganisms are italicized, and the bacterial genus is always capitalized.- The suggested correction has been implemented.
  • The methodological design of the study is missing from the Materials and Methods section (quantitative, observational, cross-sectional, etc.). – I have completed the first part of the Methods section with the following statement: “This study is a systematic review of experimental and observational studies, conducted according to the Preferred Reporting Items for Systematic Reviews and Meta-Analyses (PRISMA) guidelines"
  • All abbreviations must be described in the text the first time they are mentioned. – Thank you for this comment. The suggested correction has been implemented.
  • The tables are independent; therefore, all study information and abbreviations must be included in each table. – This matter has been resolved. I have included the descriptions of the abbreviations within the tables, either within the table titles (Table 1) or below the tables (Table 2).
  • The authors may include limitations of the study. – Thank you for this comment. The suggested correction has been implemented. A relevant statement has been added to the discussion (lines 353-36).
  • The references are not formatted uniformly and are not consistent with the journal's style. Review and correction are recommended. – The aforementioned issue has been addressed. I have updated references using the MDPI Style downloaded from the Biomedicines Website for Endnote citation manager. 

Reviewer 2 Report

Comments and Suggestions for Authors

Dear authors,

Your work is indeed very interesting and I consider that this manuscript if published would be of great interest for scientific researchers and clinicians.

You thoroughly described the available findings regarding FMT available in the literature for both adults and children suffering of different conditions.

Nevertheless, I consider that it would be very helpful for the readers if you introduced in the revised form of your manuscript a conclusion statement at the end of each section regarding the indications/potential risks or side effects of FMT for each described condition and population.

Best regards!

Author Response

We would like to express our sincere gratitude to the reviewers for their thoughtful comments, insightful suggestions, and constructive criticism. Their valuable feedback has significantly contributed to improving the quality and clarity of our manuscript. We have carefully addressed each point raised, and we believe that the revisions made have strengthened the work and enhanced its potential contribution to the medical community. We appreciate the opportunity to refine our study and hope that the revised version meets the expectations of the reviewers and editors.

To make the revised manuscript clearer, the passages that have been amended have been highlighted in yellow.

Responses to comments:

  • Nevertheless, I consider that it would be very helpful for the readers if you introduced in the revised form of your manuscript a conclusion statement at the end of each section regarding the indications/potential risks or side effects of FMT for each described condition and population.- Thank you for this comment. The suggested correction has been implemented. In order to enhance clarity, each section now concludes with a summary (lines 185-189, 237-242, 278-282)

Reviewer 3 Report

Comments and Suggestions for Authors

I have reviewed the manuscript entitled “Possible Applications of Fecal Microbiota Transplantation in the Pediatric Population: A Systematic Review.” While the topic is timely and of significant interest, I believe the paper requires substantial revision before it can be considered for publication. My key recommendations are as follows:

  1. Introduction

Please include a concise definition of Fecal Microbiota Transplantation (FMT), its therapeutic objectives, and the various modes of administration (e.g., colonoscopy, nasoenteric tube, oral capsules).

Expand the clinical applications section to cover both well-established and emerging uses of FMT.

  1. Removal of Methods Section

As this work is a retrospective review rather than an original experimental study, the detailed “Methods” section is not appropriate. I recommend deleting it and proceeding directly to the Results or main body of the review.

  1. Data Presentation

Introduce summary tables that categorize recent studies by disease, the number of publications, key outcomes, and the most recent advances.

These tables will allow readers to compare efficacy and safety across indications at a glance. Ensure both tables and narrative text remain concise.

  1. Discussion

Revise the Discussion to explicitly address the advantages and limitations of FMT in pediatric patients.

Clarify the clinical significance of this review and outline specific future research directions and knowledge gaps.

  1. Language and Style

The manuscript contains numerous non‑idiomatic English expressions and occasional grammatical errors. A thorough language edit by a native speaker or professional scientific editor is strongly recommended.

Addressing these points will strengthen the manuscript’s clarity, coherence, and scientific rigor. Accordingly, I recommend that the authors undertake a major revision before further consideration for publication.

Comments on the Quality of English Language

There are numerous instances where the English expressions do not conform to standard academic or idiomatic English usage.A thorough language revision by a native English speaker or a professional editor is highly recommended to improve clarity, fluency, and academic tone.

Author Response

We would like to express our sincere gratitude to the reviewers for their thoughtful comments, insightful suggestions, and constructive criticism. Their valuable feedback has significantly contributed to improving the quality and clarity of our manuscript. We have carefully addressed each point raised, and we believe that the revisions made have strengthened the work and enhanced its potential contribution to the medical community. We appreciate the opportunity to refine our study and hope that the revised version meets the expectations of the reviewers and editors.

To make the revised manuscript clearer, the passages that have been amended have been highlighted in yellow.

Responses to comments:

  • Introduction - Please include a concise definition of Fecal Microbiota Transplantation (FMT), its therapeutic objectives, and the various modes of administration (e.g., colonoscopy, nasoenteric tube, oral capsules). Expand the clinical applications section to cover both well-established and emerging uses of FMT. –  Thank you for this comment. The suggested correction has been implemented. I have added suitable paragraph (lines 45-67).

  • Removal of Methods Section. As this work is a retrospective review rather than an original experimental study, the detailed “Methods” section is not appropriate. I recommend deleting it and proceeding directly to the Results or main body of the review. – Thank you for this comment. According to PRISMA guidelines the Systematic Review should include the methods section, which includes the detailed process of the publication selection process. We have adhered to PRISMA guidelines in the preparation of this manuscript.

  • Data Presentation. Introduce summary tables that categorize recent studies by disease, the number of publications, key outcomes, and the most recent advances. These tables will allow readers to compare efficacy and safety across indications at a glance. Ensure both tables and narrative text remain concise. - Thank you very much for this comment. I have made every effort to ensure that the manuscript text and tables are clear and easily understandable. It was of utmost importance to me that readers could straightforwardly analyze the available studies. Therefore, I have chosen a table format that presents each study sequentially. To further improve the quality of the tables, and in accordance with the reviewer’s suggestion, the number of publications related to each disease entity presented has been added to Table 1.

  • Discussion. Revise the Discussion to explicitly address the advantages and limitations of FMT in pediatric patients. - This matter has been resolved. I have revised the manuscript and the entire discussion section has been edited to address the advantages and limitations of FMT in pediatric patients in more details. 

  • Discussion. Clarify the clinical significance of this review and outline specific future research directions and knowledge gaps. -  Thank you very much for your valuable comment. I have revised the manuscript by emphasizing key findings and trends, with the aim of identifying potential markers or therapeutic targets that could be incorporated into routine practice. This is expected to facilitate the development of more personalized and effective interventions. An appropriate paragraph addressing this has been added, covering lines 345-352.
  • Language and Style. The manuscript contains numerous non‑idiomatic English expressions and occasional grammatical errors. A thorough language edit by a native speaker or professional scientific editor is strongly recommended. -  Thank you very much for your comment. The manuscript have been thoroughly proofread and edited by a professional English translation and editing service to ensure clarity and accuracy before submission.